# Biomass and Methane Production in Double Cereal Cropping Systems with Different Winter Cereal and Maize Plant Densities

**Massimo Blandino \*, Mattia Scapino, Luca Rollè, Elio Dinuccio and Amedeo Reyneri**

Dipartimento di Scienze Agrarie, Forestali e Alimentari, Università di Torino, Largo Paolo Braccini 2, 10095 Grugliasco, Italy

\* Correspondence: massimo.blandino@unito.it; Tel.:+0039-011-6708895

**Abstract:** The biogas supply chain requires a correct combination of crops to maximize the methane yield per hectare. Field trials were carried out in North Italy over three growing seasons, according to a factorial combination of four cropping systems (maize as a sole-crop or after hybrid barley, triticale and wheat) and two maize plant densities (standard, 7.5 plants m$^{-2}$ and high, 10 plants m$^{-2}$) with the plants harvested as whole-crop silage. The specific methane production per ton was measured through the biochemical methane potential (BMP) method, while the methane yield per hectare was calculated on the basis of the BMP results and considering the biomass yield. The average methane yield of wheat resulted to be equal to 4550 Nm$^3$ ha$^{-1}$, and +17% and +28% higher than triticale and barley, respectively, according to the biomass yield. A delay in maize sowing reduced the yield potential of this crop; the biomass of maize grown after barley, triticale and wheat was 20%, 33% and 47% lower, respectively, than maize cultivated as a single crop. The high plant population increased the biomass yield in the sole-crop maize (+23%) and in the maize grown after barley (+20%), compared to the standard density. The highest biomass (32 t ha$^{-1}$ DM) and methane yield (9971 Nm$^3$ ha$^{-1}$) within the cropping systems were obtained for barley followed by maize at a high plant density. This cropping system increased the methane yield by 46% and 18%, respectively, compared to the sole-crop maize or maize after triticale at a standard density. The smaller amount of available solar radiation, resulting from the later sowing of maize, reduced the advantage related to the application of a high plant density.

**Keywords:** biogas; hybrid barley; triticale; wheat; maize; plant population

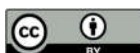

## 1. Introduction

Anaerobic digestion is a well-established technology used for the production of renewable energy from agricultural biomasses. European countries are leaders in the world's agricultural biogas production, with a combined biogas and biomethane production in 2021 that amounted to 196 TWh of energy, from 14.90 billion cubic meters (bcm) of biogas and 3.50 bcm of biomethane [1]. Interest in anaerobic digestion applications has grown over the past few years, especially concerning the utilization of manure and agroindustry waste as primary biomass sources for biogas production [2–4]. Nonetheless, most plants still rely on dedicated energy crops as their main or secondary biomass source [5–7]. This is due to the role energy crops play in achieving higher methane yields and in stabilizing biogas production [8]. Furthermore, the use of energy crops to feed biogas plants results in competition in the exploitation of land for food, feed and energy purposes [8]. This competition has led to the need to find new cropping systems that are more efficient and competitive per area unit, to enhance the methane yield per hectare and reduce the use of soil for energy purposes. Several optimization approaches that have focused on different agronomical management strategies, such as double cropping, the recovery of crop residues and the introduction of alternative-innovative crops, have been proposed

[9–11]. It is well known that double cropping increases annual productivity, due to its more complete use of resources than single crops [12]. In temperate conditions, the crops most commonly used for anaerobic digestion are maize (*Zea mays* L.), winter cereals and grasses, harvested as green whole-crops for silage forage [13]. Maize biomass is one of the main components used for anaerobic digestion, since it has the highest yield potential of all the crops cultivated in Central and South Europe [14], and it offers a particularly suitable substrate for anaerobic fermentation [15], due to its high starch content and highly digestible fiber [16]. However, the introduction of alternative crops to maize in irrigated temperate growing areas characterized by high yield potential, is not sufficient to provide the same benefits, in terms of overall biogas yield per hectare [17,18]. In these growing areas, the conventional cropping system generally adopted by farmers to feed biogas plants involves the insertion of maize, chopped at the dough stage, into a double cropping system in combination with a winter cereal, such as triticale (× *Triticosecale* Wittmack) harvested at the early dough stage [19]. In these production situations, triticale allows a satisfactory biomass to be obtained, in terms of quantity and quality, thus providing a good response to fertilization with organic N and catching nitrogen leaching from the soil during autumn and winter [9,20,21]. Garuti et al. [22] reported that the cellulose content and degradability mainly affect the methane production of different triticale cultivars. Furthermore, to increase both the sustainability and profitability of intensive cereal cropping systems for methane production, it is important to introduce genetic and agronomic innovations that are able to enhance their productivity and efficiency, without any significant increase in the cultivation costs. Optimizing the planting density of maize is considered an efficient way of improving the interception of active radiation [23], and therefore, of achieving higher yields with similar inputs. Testa et al. [24] observed a significant grain yield enhancement for a full-season maize hybrid cultivated in North Italy, when the plant density was increased to 10 plants m$^{-2}$; this was achieved by reducing the inter-row spacing from the conventional 0.75 m to a narrower 0.5 m, which led to a better optimization of the plant distribution in the field and an overall higher light interception. Similar benefits could also be expected for the biomass yield of silage maize, especially since there are no potential limitations associated with the dry down process. F$_1$ hybrid barley (*Hordeum vulgare* L.) varieties, which are characterized by a higher stay-green and biomass yields than conventional (no hybrid) barley cultivars, represent a recent introduction of a winter crop used for biomass purposes, because of the exploitation of heterosis associated with the hybridization process [25–27]. As a result of its more rapid crop development, hybrid barley can be harvested earlier than triticale and this would allow an earlier sowing of maize in rotation to be introduced, thus favoring its development because of potential higher solar radiation availability. Another winter cereal that can be used, albeit through a different approach, to feed biogas plants is that of common wheat (*Triticum aestivum* spp. *aestivum* L.), a crop that can easily be inserted in rotation with summer cereals and which is able to ensure a higher yield, in terms of starch and protein, than triticale, in addition to high dry matter (DM) digestibility [28,29]. Regarding silage, winter wheat is interesting because of its potential quality–quantity characteristics when it is used as a fermented whole-crop harvested at an early dough stage, which leads to a higher DM intake and improved rumen fermentation in cows [29,30]. Recent experimental studies have underlined the energetic advantages of the use of recent wheat cultivars specifically developed for the production of biomass and featuring a high size of the plant and high stay green [31]. By comparing the methane yield of a wide range of crops, Hermann et al. [32] reported that winter barley, triticale and wheat showed similar methane potential, although as far as the chemical composition of silage is concerned, the protein content of barley and the starch content of wheat were slightly higher than triticale.

In order to maximize the biomass yield in methane per hectare, the aim of this study has been to compare maize cropping systems that adopt conventional or high-density sowing strategies, and to consider their combination with different cereal crops, such as F$_1$ hybrid barley, wheat and triticale, which are characterized by different precocities.

## 2. Materials and Methods

### 2.1. Experimental Site and Treatments

Large-scale field trials were carried out over 3 growing seasons, from 2013 to 2016, in Northwest Italy. The experiments were carried out in Candiolo (44°57′36″72 N, 07°36′10″08 E) over the 2013–14 season, and in Carignano (44°50′52″80 N, 07°43′8″76 E) over the 2014–15 and 2015–16 growing seasons. The treatments compared in each trial were factorial combinations, in the same field, of:

- Four cropping systems based on single or double crops harvested as whole-crop silage:
  - M, sole-crop maize planted in spring,
  - BM, double cropping system with hybrid barley followed by maize as an intercrop,
  - TM, double cropping system with triticale followed by maize as an intercrop,
  - WM, double cropping system with common wheat followed by maize as an intercrop.
- Two plant densities for the maize crop:
  - StD, a standard planting density (7.5 plants m$^{-2}$) with plants sown at a 0.75 m wide inter-row spacing, and an average distance of 0.18 m between two contiguous plants,
  - HiD, a high planting density (10 plants m$^{-2}$) with plants sown at a narrow inter-row spacing of 0.5 m, and a distance between plants of 0.2 m in the same row.

The treatments were assigned to experimental units using a split-plot design, with the cropping system as the main-plot treatment and the maize plant density as the sub-plot treatment. The experimental unit was replicated 3 times on a surface that was able to allow the crop to be harvested with a conventional chopping machine. The 150 m long sub-plot consisted of 8 rows and 12 rows for the StD and HiD planting systems, respectively. All the compared winter cereals were sown at the same time, between the end of October and the beginning of November, while the harvest was performed at an early dough stage of the grain (growth stage GS 81–83, [33]) with approximately 7 days of interval between the crops, according to their precocity (barley > triticale > wheat). The maize in the single cropping system was sown at the beginning of April and harvested at the end of August, after a visual assessment of the kernel milk-line position at the dough stage (GS85; [34]). The maize in the double cropping systems was instead sown a few days after the harvest of the previous winter cereals (with an interval of approximately 5–7 days) and harvested at the end of September or the beginning of October, when the right growth stage for silage production had been reached. The main physical and chemical parameters of each experiment are reported in Table 1. The sowing and harvesting dates of each crop are reported in Table 2 for all the considered growing seasons.

The compared winter cereals were the Volume (Syngenta Italia S.p.A., Milano, Italy), Tarzan (Società Italiana Sementi S.p.A, San Lazzaro di Savena, Bologna, Italy), and Illico (Syngenta Italia S.p.A., Milano, Italy) cultivars for F$_1$ crossbreed hybrid barley, conventional triticale and conventional common wheat, respectively. All the considered genotypes have a dual-purpose (grain and biomass) aptitude. The triticale and wheat cultivars used in the study were pure lines (no F$_1$ hybrid). Studies were carried out on a commercial SY Hydro dent maize hybrid (FAO maturity class 600, 135 days relative to maturity; Syngenta Italia S.p.A., Milano, Italy) in Candiolo in the 2013–14 period and on the SY Sincero hybrid (FAO maturity class 500, 127 days relative to maturity; Syngenta Italia S.p.A., Milano, Italy) in Carignano in the 2014–15 and 2015–16 growing seasons. The previous crop in all the experimental fields was maize for silage, which was cultivated as a sole-crop system. Planting was carried out for all the compared winter cereals and for the single maize cropping system after an autumn 0.3 m deep ploughing, followed by disk harrowing, to prepare a suitable seedbed. The maize that followed the winter cereals was instead

sown after a minimum tillage, through a double disk and rotary harrowing, in order to facilitate a rapid sowing after the previous crop harvest.

**Table 1.** Main physical and chemical characteristics of the experimental site.

| Parameters | Measurement Units | Candiolo | Carignano |
|---|---|---|---|
| Sand (2–0.05 mm) | % | 54.7 | 35.5 |
| Silt (0.05–0.002 mm) | % | 5.5 | 57.9 |
| Clay (<0.002 mm) | % | 1.9 | 6.6 |
| pH | | 8.0 | 8.1 |
| Organic matter | % | 1.9 | 2.3 |
| C/N | | 10.1 | 8.9 |
| Cation Exchange Capacity (C.E.C.) | meq 100 $g^{-1}$ | 9.2 | 12.0 |
| N | % | 0.13 | 0.15 |
| Available $P_2O_5$ | mg $kg^{-1}$ | 22 | 21 |
| Exchangeable K | mg $kg^{-1}$ | 201 | 174.0 |

**Table 2.** Sowing and harvesting dates of the cropping systems in the three growing seasons.

| Growing Season | Crop | Sowing Date | Harvesting Date |
|---|---|---|---|
| 2013–14 | Barley | 30 October 2013 | 26 May 2014 |
| | Triticale | 30 October 2013 | 4 June 2014 |
| | Wheat | 30 October 2013 | 13 June 2014 |
| | Maize | 1 April 2014 | 21 August 2014 |
| | Maize after barley | 5 June 2014 | 7 October 2014 |
| | Maize after triticale | 11 June 2014 | 7 October 2014 |
| | Maize after wheat | 23 June 2014 | 7 October 2014 |
| 2014–15 | Barley | 3 November 2014 | 27 May 2015 |
| | Triticale | 3 November 2014 | 8 June 2015 |
| | Wheat | 3 November 2014 | 17 June 2015 |
| | Maize | 1 April 2015 | 5 August 2015 |
| | Maize after barley | 3 June 2015 | 29 September 2015 |
| | Maize after triticale | 10 June 2015 | 13 October 2015 |
| | Maize after wheat | 19 June 2015 | 13 October 2015 |
| 2015–16 | Barley | 27 October 2015 | 30 May 2016 |
| | Triticale | 27 October 2015 | 11 June 2016 |
| | Wheat | 27 October 2015 | 17 June 2016 |
| | Maize | 1 April 2016 | 19 August 2016 |
| | Maize after barley | 31 May 2016 | 12 September 2016 |
| | Maize after triticale | 14 June 2016 | 12 September 2016 |
| | Maize after wheat | 24 June 2016 | 5 October 2016 |

The maize in all the sites was irrigated using the border flooding surface method to maintain the water-holding capacity at between 33 and 200 kPa. The single maize cropping system received approximately 300, 150 and 200 kg $ha^{-1}$ of N, $P_2O_5$ and $K_2O$, respectively, in each growing season through the distribution of the digestate before spring harrowing and urea distribution at the 8th leaf stage (GS28). The double-cropping system overall received 450, 225 and 300 kg $ha^{-1}$ of N, $P_2O_5$ and $K_2O$, respectively, through the distribution of the digestate, which was split into an autumn ploughing, before winter cereal sowing, and a summer harrowing, before maize sowing. The fertilizer rate was the same for all the compared winter cereals and for both maize density treatments. The other agronomical practices (weed and pest control) were conducted according to the ordinary

agronomic techniques adopted in the cultivation area and were the same for all the compared winter cereals and for both maize density treatments.

*2.2. Biomass and Methane Yield and Qualitative Measurements*

The biomass yield was obtained after chopper harvesting of the whole plots at an appropriate maturity stage for the silage of each crop and by weighing the biomass harvested from the entire plot surface. The harvest was carried out using a self-propelled forage harvester (Class Jaguar 960, Harsewinkel, Westphalia, Germany), equipped with 2 counter-rotating rolls and set at a 19-mm theoretical chop length. About 1 kg of chopped fresh sample from each plot was weighed before and after being dried at 120 °C until constant weight to assess the DM content. Another 2 kg subsample was collected, dried at 65 °C for 48 h and milled to 0.500 mm to establish the chemical composition. The crude protein, starch, neutral detergent fiber (NDF), acid detergent fiber (ADF) and acid detergent lignin (ADL) contents were analyzed using a near infrared instrument (NIR system 5000 FOSS®) [35]. Moreover, 4 kg representative samples of fresh chopped whole plants were taken at harvest and stored in a refrigerator at a constant temperature (4 °C) under vacuum for a maximum of 2 weeks before biochemical methane potential (BMP) analysis. The BMP tests were performed according to VDI 4630 [36] and following the experimental procedure described by Dinuccio et al. [37]. Two-liter capacity batch reactors were filled with a mixture of feedstock, inoculum and deionized water to obtain a final feedstock-to-inoculum ratio of 1:2, on the basis of the content of the volatile solids (VS). The used inoculum consisted of the separate liquid fraction of digested slurry produced in an anaerobic digester plant fed with a mixture of pig slurry (70%), farmyard manure (4%), maize silage (14%) and sorghum silage (12%). Blank batch trials were also carried out with only inoculum; the biogas residual potential was measured and subtracted from the biogas obtained from the samples. Trials were performed in a temperature-controlled incubator under mesophilic conditions (40 ± 2 °C) for a period of 60 days. Each biomass was digested in triplicate. The produced biogas was collected in Tedlar® bags, and its volume and composition were monitored every 3 days for the first 2 weeks, and then weekly until the end of the test. The biogas volume was measured by means of a Ritter Drum-type Gas volume meter (TG05/5, Ritter Apparatebau GmbH & Co. KG, Bochum, Germany). The methane concentration in the biogas was determined using a gas analyzer equipped with infrared sensors (model XAM 7000, Drägerwerk AG & Co. KgaA, Lübeck, Germany). The recorded data were normalized to the standard temperature and pressure (0 °C and 1013 hPa), according to VDI 4630 [36]. The results were expressed as potential methane production per hectare.

*2.3. Statistical Analysis*

The normal distribution and homogeneity of variance were verified by performing the Kolmogorov–Smirnov normality and Levene test. An analysis of variance (ANOVA) was run separately for each growing season, to analyze the effect of the single crops and their combination in different cropping system on the biomass yield and composition, biochemical methane potential (BMP) and methane yield per hectare. When necessary, post hoc multiple comparison tests were performed, by means of the Ryan–Einot–Gabriel–Welsh Q (REGW-Q) test. SPSS Version 24 for Windows statistical package (SPSS Inc., Chicago, IL, USA) was used for the statistical analysis.

**3. Results**

*3.1. Meteorological Trends*

The three growing seasons were subject to different meteorological trends, as far as both rainfall and temperature are concerned (Table 3).

**Table 3.** Total rainfall, rainy days and growing degree days (GDDs) [1] in the three growing seasons.

| Growing Season | Month | Rainfall (mm) | Rainy Days (n°) | GDDs Cereal (Σ °C-Day) | GDDs Maize (Σ °C-Day) |
|---|---|---|---|---|---|
| 2013–14 | November | 96.8 | 16 | 264 | |
| | December | 65.8 | 15 | 168 | |
| | January | 66.0 | 15 | 157 | |
| | February | 86.6 | 18 | 177 | |
| | March | 88.8 | 11 | 318 | |
| | April | 60.2 | 10 | 436 | 169 |
| | May | 97.4 | 14 | 524 | 242 |
| | June | 67.4 | 14 | | 339 |
| | July | 89.6 | 18 | | 364 |
| | August | 73.6 | 12 | | 370 |
| | September | 50.0 | 11 | | 286 |
| | October | 22.4 | 13 | | 136 |
| | November–May | 562 | 99 | 2044 | |
| | May–October | 303 | 68 | | 1496 |
| 2014–15 | November | 271.0 | 15 | 289 | |
| | December | 92.2 | 9 | 185 | |
| | January | 35.6 | 10 | 139 | |
| | February | 205.6 | 11 | 143 | |
| | March | 188.4 | 9 | 300 | |
| | April | 66.6 | 7 | 416 | 171 |
| | May | 86.0 | 11 | 581 | 273 |
| | June | 55.0 | 7 | | 359 |
| | July | 9.2 | 3 | | 446 |
| | August | 71.8 | 8 | | 379 |
| | September | 52.4 | 5 | | 260 |
| | October | 147.6 | 13 | | 137 |
| | November–May | 945 | 72 | 2054 | |
| | May–October | 336 | 36 | | 1580 |
| 2015–16 | November | 3.2 | 1 | 284 | |
| | December | 2.0 | 1 | 182 | |
| | January | 4.4 | 4 | 159 | |
| | February | 127.6 | 10 | 190 | |
| | March | 70.6 | 6 | 286 | |
| | April | 79.8 | 9 | 428 | 163 |
| | May | 112.0 | 18 | 517 | 222 |
| | June | 36.6 | 13 | | 336 |
| | July | 17.8 | 7 | | 412 |
| | August | 5.4 | 3 | | 396 |
| | September | 24.6 | 8 | | 309 |
| | October | 59.6 | 7 | | 122 |
| | November–May | 400 | 49 | 2047 | |
| | May–October | 144 | 38 | | 1575 |

Source: Rete Agrometeorologica del Piemonte-Regione Piemonte-Assessorato Agricoltura-Settore Fitosanitario, sezione di Agrometeorologia. [1] Accumulated growing degree days for each experiment using a 0 °C base value for winter cereals and 10 °C base for maize.

The highest level of precipitation was recorded during the 2014–15 season, and it was mainly concentrated during the winter and spring months. The rainfall was lower in summer (June–August) in both the 2014–15 and 2015–16 growing seasons, while the precipitation was better distributed over the 2013–14 period. The Growing Degree Days (GDD)

for winter cereal development from November to May were similar for all the considered growing seasons. However, 2014 had fewer GDDs for maize from June to October than in 2015 and 2016.

### 3.2. Winter Cereal Biomass and Methane Yield

The barley harvested at the early dough stage was on average anticipated by −7 days and −17 days, compared to triticale and wheat, respectively (Table 1). The differences between the winter cereal crops were significant for the biomass yield in all the growing seasons (Table 4). Wheat resulted in the highest biomass production of the winter cereals: on average this crop had 30.6% and 22.1% higher yield values than barley and triticale, respectively. The difference in the hybrid barley and triticale biomass was not significant in the 2013–14 and 2014–15 growing seasons, while only in 2015–16 did triticale have a 12.6% greater biomass yield than barley.

**Table 4.** Biomass yield, biochemical methane potential (BMP) and methane yield per hectare of the considered winter cereals [1].

| Growing Season | Winter Cereal | Biomass Yield (t ha⁻¹ DM) | DM (%) | VS (% DM) | BMP (Nm³CH₄ t VS⁻¹) | Methane Yield (Nm³ ha⁻¹) |
|---|---|---|---|---|---|---|
| 2013–14 | barley | 12.0 b | 21.4 c | 90.6 a | 335 a | 3634 b |
|  | triticale | 12.5 b | 27.3 b | 91.8 a | 346 a | 3968 b |
|  | wheat | 16.0 a | 35.5 a | 90.9 a | 331 a | 4811 a |
|  | $P$ (F) | * | ** | ns | ns | *** |
| 2014–15 | barley | 11.7 b | 27.9 c | 89.6 a | 342 a | 3650 b |
|  | triticale | 12.1 b | 38.9 b | 90.3 a | 348 a | 3828 b |
|  | wheat | 14.0 a | 49.0 a | 90.7 a | 339 a | 4368 a |
|  | $P$ (F) | ** | *** | ns | ns | ** |
| 2015–16 | barley | 10.6 c | 26.8 b | 90.2 a | 355 a | 3412 b |
|  | triticale | 12.0 b | 34.9 a | 92.3 a | 350 a | 3871 b |
|  | wheat | 14.7 a | 34.3 a | 89.2 a | 341 a | 4470 a |
|  | $P$ (F) | *** | ** | ns | ns | * |

[1] see Table 2 for details of the harvesting times; DM, dry matter; VS, volatile solids; BMP, Biochemical Methane Potential expressed per t of VS of maize. Means followed by different letters are significantly different, according to the REGW-Q test. Level of significance: * $p \leq 0.05$; ** $p \leq 0.01$; *** $p \leq 0.001$, ns: not significant.

The DM content was significantly lower for hybrid barley than for triticale and wheat in all the growing seasons. No significant differences were observed for the volatile solid concentration between the winter cereals, while the chemical composition of the whole plant at harvest varied between the crops across the considered growing seasons (Table 5). In the 2013–14 and 2014–15 growing seasons, triticale resulted in a lower starch and higher NDF content than barley and wheat, while the compared winter cereals showed similar fiber and starch concentrations in the 2015–16 period. Wheat resulted in higher ADL values than barley and triticale in the 2013–14 and 2014–15 growing seasons. As far as the protein content is concerned, no differences were reported for 2013–14, while the concentration of crude protein was higher in the hybrid barley and wheat in the 2014–15 and 2015–16 growing seasons. Despite the previous variability in biomass composition, no significant differences were reported for BMP for the compared crops (Table 6).

**Table 5.** Biomass composition of the three winter cereals [1].

| Growing Season | Winter Cereal | NDF (% DM) | ADF (% DM) | ADL (% DM) | Starch (% DM) | Crude Protein (% DM) |
|---|---|---|---|---|---|---|
| 2013–14 | barley | 55.3 b | 36.3 b | 4.4 b | 12.7 a | 8.8 a |
| | triticale | 58.5 a | 40.6 a | 4.6 b | 5.8 b | 7.1 a |
| | wheat | 56.1 b | 39.9 a | 5.2 a | 18.2 a | 7.6 a |
| | *P* (F) | ** | * | * | * | ns |
| 2014–15 | barley | 54.6 a | 38.0 b | 5.3 a | 16.2 a | 8.8 a |
| | triticale | 56.9 a | 48.4 a | 6.2 a | 8.7 b | 5.9 c |
| | wheat | 55.8 a | 37.3 b | 5.9 a | 19.0 a | 7.4 b |
| | *P* (F) | ns | * | ns | * | * |
| 2015–16 | barley | 53.5 a | 36.3 ab | 5.4 ab | 10.4 a | 8.5 b |
| | triticale | 51.8 a | 34.1 b | 5.0 b | 14.2 a | 9.3 ab |
| | wheat | 53.7 a | 38.0 a | 5.9 a | 15.0 a | 10.0 a |
| | *P* (F) | ns | * | * | ns | * |

[1] see Table 2 for details of the harvesting times. DM, dry matter; NDF, neutral detergent fiber; ADF, acid detergent fiber; ADL, acid detergent lignin. Means followed by different letters are significantly different, according to the REGW-Q test. Level of significance: * $p \leq 0.05$; ** $p \leq 0.01$; ns: not significant.

As for the methane yield per hectare, wheat always resulted in the highest value: on average this crop had a 28% and 17% higher value than barley and triticale, respectively. No significant differences between hybrid barley and triticale were observed for any of the growing seasons.

### 3.3. Maize Biomass and Methane Yield

ANOVA showed a significant effect of the combination of the maize sole crop/intercrop and plant density on the biomass yield for all the growing seasons (Table 6). As expected, the delay in maize sowing reduced the biomass production: compared to the maize cultivated as a sole-crop, the yield was on average −20%, −33% and −47% for maize after barley, triticale and wheat, respectively. In the cropping system with sole-crop maize, the HiD significantly increased the biomass yield by 33%, 20% and 16%, compared to StD, in the 2013–14, 2014–15 and 2015–16 growing season, respectively. A significant advantage of applying the HiD system was also observed for maize after barley (+32%) and triticale (+26%) in the growing season with the best distributed rainfall during the summer (2013–14), while, in 2015–16, only the HiD maize that followed barley resulted in a biomass yield advantage (+14%) over the StD one. In the driest growing season during summer (2014–15), HiD maize did not lead to a significant increase in the biomass yield, compared to StD, for any of the sowing times after a winter cereal crop. The increase in the plant population (HiD) in maize sown after wheat did not lead to any significant increase in the biomass yield for any of the considered growing seasons. Overall, the HiD maize sown in early spring (single crop) resulted in a significantly higher biomass yield than the other maize cropping systems in all the growing seasons. In each growing seasons, the DM content resulted higher for the early-planted maize. The VS content was only affected to a great extent by the maize cropping systems in the 2013–14 growing season, and showed greater values for the single-crop maize. The fiber fraction, starch and protein contents generally resulted to be steady for the maize over the different cropping systems (Table 7).

**Table 6.** Biomass yield, biochemical methane potential (BMP), and methane yield per hectare of the maize crops for different sowing times [1] and plant densities [2].

| Growing Season | Maize | Plant Density | Biomass Yield (t ha⁻¹ DM) | DM (%) | VS (% DM) | BMP (Nm³CH₄ t VS⁻¹) | Methane Yield (Nm³ ha⁻¹) |
|---|---|---|---|---|---|---|---|
| 2013–14 | single crop | StD | 23.0 c | 30.8 ab | 91.3 a | 326 a | 6838 bc |
| | after barley | StD | 19.6 de | 32.0 a | 88.1 ab | 309 a | 5357 d |
| | after triticale | StD | 16.3 f | 27.8 abc | 83.3 cd | 346 a | 4698 d |
| | after wheat | StD | 17.6 ef | 23.7 bc | 88.0 abc | 316 a | 4905 d |
| | single crop | HiD | 30.6 a | 28.1 abc | 90.7 a | 332 a | 9192 a |
| | after barley | HiD | 25.9 b | 32.7 a | 85.0 bcd | 336 a | 7405 b |
| | after triticale | HiD | 20.5 de | 29.2 abc | 80.2 d | 385 a | 6323 c |
| | after wheat | HiD | 17.7 ef | 22.2 c | 88.2 ab | 326 a | 5073 d |
| | *P* (F) | | *** | *** | *** | ns | *** |
| 2014–15 | single crop | StD | 21.0 b | 28.5 cd | 95.7 a | 402 a | 8102 a |
| | after barley | StD | 16.1 cd | 34.5 ab | 95.1 a | 306 a | 4701 bc |
| | after triticale | StD | 13.3 ef | 34.5 ab | 96.0 a | 317 a | 4038 bc |
| | after wheat | StD | 10.1 f | 29.1 cd | 95.3 a | 339 a | 3266 c |
| | single crop | HiD | 25.2 a | 25.9 d | 95.2 a | 394 a | 9440 a |
| | after barley | HiD | 18.4 c | 37.9 a | 96.1 a | 316 a | 5587 b |
| | after triticale | HiD | 14.7 de | 32.0 bc | 95.1 a | 331 a | 4597 bc |
| | after wheat | HiD | 9.9 f | 32.0 bc | 95.8 a | 327 a | 3113 c |
| | *P* (F) | | *** | *** | ns | ns | *** |
| 2015–16 | single crop | StD | 19.8 b | 33.6 a | 95.4 a | 334 a | 6311 b |
| | after barley | StD | 15.9 c | 32.7 ab | 96.2 a | 352 a | 5390 c |
| | after triticale | StD | 15.1 c | 29.8 bc | 95.6 a | 349 a | 5030 c |
| | after wheat | StD | 10.4 de | 27.9 c | 95.3 a | 338 a | 3342 d |
| | single crop | HiD | 23.1 a | 30.1 bc | 95.1 a | 338 a | 7414 a |
| | after barley | HiD | 18.2 b | 29.3 c | 96.3 a | 355 a | 6223 b |
| | after triticale | HiD | 15.8 c | 31.3 abc | 94.8 a | 320 a | 4793 c |
| | after wheat | HiD | 10.8 de | 28.0 c | 95.6 a | 344 a | 3558 d |
| | *P* (F) | | *** | *** | ns | ns | *** |

[1] see Table 2 for details of the sowing and harvesting times. [2] StD, a standard planting density (7.5 plants per m⁻²) sown at a wide inter-row spacing of 0.75 m; HiD, a high planting density (10 plants per m⁻²) with a narrow inter-row spacing of 0.5 m. DM, dry matter; VS, volatile solids; BMP, Biochemical Methane Potential expressed per t of VS of maize. Means followed by different letters are significantly different, according to the REGW-Q test. Level of significance: *** $p \leq 0.001$, ns: not significant.

Significant differences for all these composition parameters were only detected in the 2013–14 growing seasons, with lower starch and high NDF, ADF and ADL contents in the late-planted maize (after triticale and wheat) than in the earlier one (maize as a sole-crop or as an intercrop after barley). As far as the BMP is concerned, no significant difference between the maize crops was observed for the compared cropping systems (Table 6). Thus, on the basis of the biomass, the methane yield per hectare of maize decreased according to the time of sowing. No significant difference between HiD and StD was detected for the plant density effect in the 2014–15 growing season. The HiD maize instead resulted in a significantly higher methane yield than StD (+17% and +15%) in the 2015–16 period for maize cultivated as a sole-crop or as an intercrop after barley, respectively. In the 2013–14 growing seasons, the methane yield was significantly increased by the HiD system, by 34%, 38% and 35% for maize cultivated as a sole-crop or an intercrop after barley and triticale, respectively.

**Table 7.** Biomass composition of the maize crop with different sowing times [1] and plant densities [2].

| Growing Season | Maize | Plant Density | NDF (% DM) | ADF (% DM) | ADL (% DM) | Starch (% DM) | Crude Protein (% DM) |
|---|---|---|---|---|---|---|---|
| 2013–14 | single crop | StD | 40.5 cd | 25.2 bc | 3.0 | 30.4 | 7.3 bc |
| | after barley | StD | 45.9 bcd | 27.8 ab | 3.0 | 31.8 | 6.0 e |
| | after triticale | StD | 51.0 ab | 31.7 ab | 3.6 | 25.1 | 6.8 bcd |
| | after wheat | StD | 48.8 abc | 30.0 ab | 3.4 | 25.1 | 7.4 b |
| | single crop | HiD | 41.9 cd | 25.5 bc | 2.8 | 30.8 | 7.2 bc |
| | after barley | HiD | 38.9 d | 22.9 c | 2.7 | 37.3 | 6.6 cde |
| | after triticale | HiD | 49.2 abc | 30.8 ab | 3.5 | 26.9 | 6.3 de |
| | after wheat | HiD | 54.6 a | 35.1 a | 4.3 | 20.0 | 8.5 a |
| | *P* (F) | | ** | ** | ** | ** | *** |
| 2014–15 | single crop | StD | 36.5 a | 22.6 a | 3.4 | 31.3 | 8.3 abc |
| | after barley | StD | 39.4 a | 22.1 a | 3.4 | 32.5 | 8.7 a |
| | after triticale | StD | 37.4 a | 21.1 a | 2.9 | 34.3 | 7.0 c |
| | after wheat | StD | 41.5 a | 24.1 a | 3.2 | 28.6 | 7.7 abc |
| | single crop | HiD | 37.8 a | 22.8 a | 3.3 | 31.6 | 8.5 ab |
| | after barley | HiD | 39.1 a | 21.8 a | 3.2 | 31.2 | 8.2 abc |
| | after triticale | HiD | 36.9 a | 21.2 a | 3.0 | 34.8 | 7.1 bc |
| | after wheat | HiD | 39.0 a | 22.8 a | 3.3 | 29.2 | 8.0 abc |
| | *P* (F) | | ns | ns | ns | ns | * |
| 2015–16 | single crop | StD | 35.1 a | 22.7 a | 3.1 | 34.0 | 8.2 a |
| | after barley | StD | 40.8 a | 26.3 a | 3.1 | 29.6 | 7.3 a |
| | after triticale | StD | 42.4 a | 27.8 a | 3.5 | 26.5 | 6.9 a |
| | after wheat | StD | 41.5 a | 24.0 a | 3.2 | 28.6 | 7.7 a |
| | single crop | HiD | 37.7 a | 25.1 | 3.3 | 33.0 | 7.9 a |
| | after barley | HiD | 44.1 a | 28.6 | 3.4 | 25.4 | 7.4 a |
| | after triticale | HiD | 41.9 a | 26.2 | 3.5 | 27.8 | 7.1 a |
| | after wheat | HiD | 39.0 a | 22.8 | 3.2 | 29.2 | 8.0 a |
| | *P* (F) | | ns | ns | ns | ns | ns |

[1] see Table 2 for details of the sowing and harvesting times. [2] StD, a standard planting density (7.5 plants per m$^{-2}$) sown at a wide inter-row spacing of 0.75 m; HiD, a high planting density (10 plants per m$^{-2}$) with a narrow inter-row spacing of 0.5 m. DM, dry matter; NDF, neutral detergent fiber; ADF, acid detergent fiber; ADL, acid detergent lignin. Means followed by different letters are significantly different, according to the REGW-Q test. Level of significance: * $p \leq 0.05$; ** $p \leq 0.01$; *** $p \leq 0.001$, ns: not significant.

### 3.4. Biomass and Methane Yield of the Cropping Systems

The effect of the whole cropping system, and of the combination of winter cereal and maize, on the biomass and methane yield per hectare is reported in Figure 1. The double cropping systems in the StD maize cultivation conditions always resulted in a significant increase in the biomass yield, compared to the system with maize as a sole-crop.

The barley–HiD maize double cropping system resulted in the highest biomass and methane yield for all the growing seasons. This treatment showed an increase in methane per hectare of 46% and 18%, compared to StD maize alone and triticale + StD maize, respectively. The 2014–15 growing season reported the lower benefit in term of methane yield of the double cropping system compared to the sole-crop maize: a significant difference for the methane yield was reported between maize cultivated as a sole-crop and barley + maize double cropping system (with both maize plant density) and between maize cultivated as a sole-crop and triticale + maize double cropping system (HiD plant density).

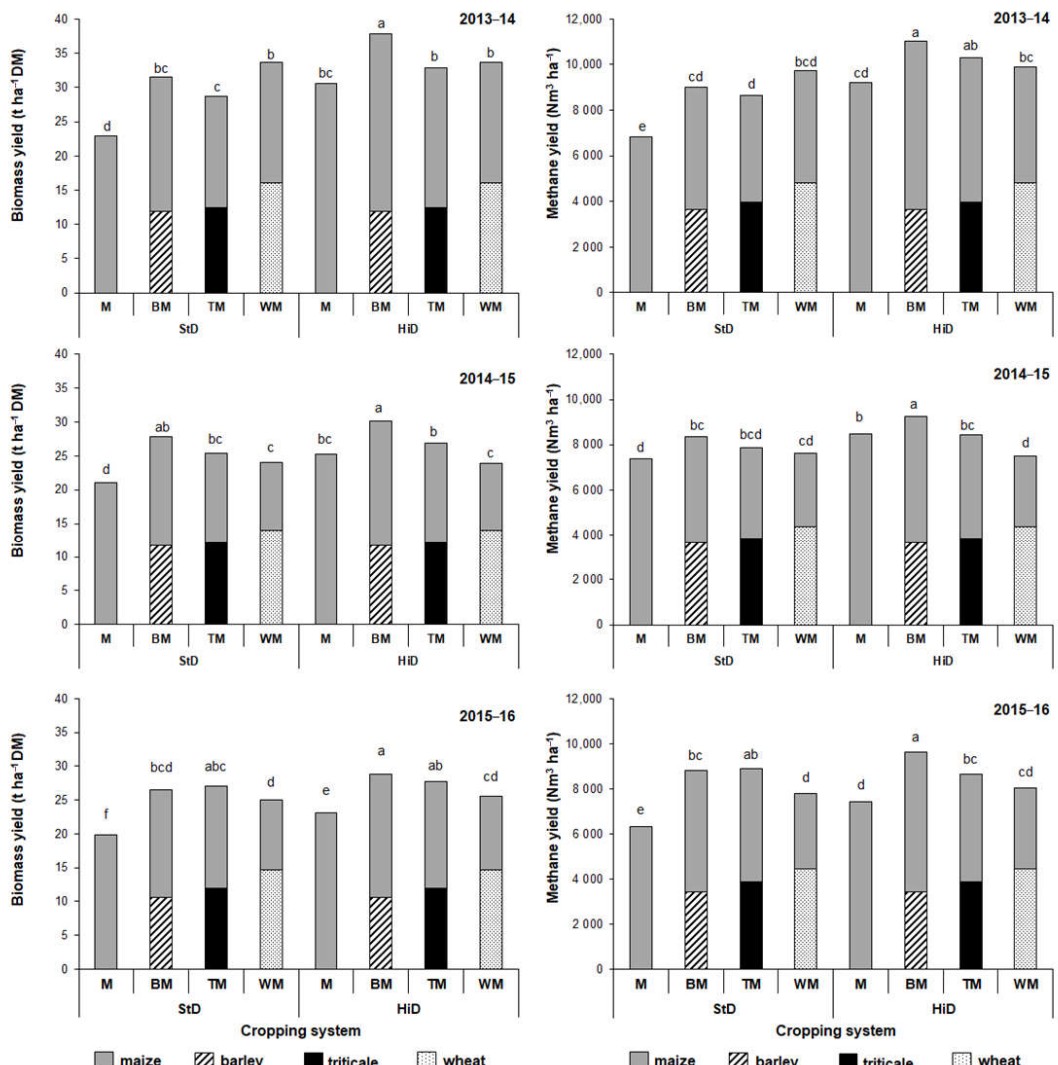

**Figure 1.** Effect of the cropping systems according to the different winter cereal-maize combinations and maize plant densities on the biomass and methane yields per hectare. M, single maize crop planted in spring; BM, double crop with hybrid barley followed by maize; TM, double crop with triticale followed by maize; WM, double crop with common wheat followed by maize. Maize plant densities: StD, a standard planting density (7.5 plants per m$^{-2}$) with plants sown at a wide inter-row spacing of 0.75 m; HiD, a high planting density (10 plants per m$^{-2}$) with plants sown at a narrow inter-row spacing of 0.5 m. Different letters on the bars indicate significant differences ($p < 0.01$), according to the REGW-Q test.

As far as the maize cultivated as a single crop is concerned, HiD always led to a significant increase in the biomass (+23%) and methane yields (+22%), compared to StD. The cropping system with HiD sole-crop maize resulted in a similar biomass and methane yield to that of the triticale–StD maize and the wheat–StD maize systems in the 2013–14 and 2014–15 growing seasons. Moreover, the application of a high plant density cultivation of maize did not lead to any significant increase in the methane yield, compared to the standard density in the triticale + maize and wheat + maize double cropping systems.

## 4. Discussion

This study further confirms that energy yields in cereal cropping systems can be implemented through a rational combination of double crops, to maximize their overall biomass production, as previously highlighted in different growing areas and environments [9,38].

Among the tested winter cereals, wheat has produced more methane per unit area than both barley and triticale. The recorded biomass advantages, as well as the differences in the harvesting date between wheat and triticale are in agreement with the findings of Mühleisen et al. [26] and Nadeau [39]. The higher biomass production of wheat is linked to a higher plant development and a longer growth period, with a harvesting date that is ~7 and ~17 days later than triticale and barley, respectively. Moreover, although the BMP of the wheat in the present experiment did not differ from those of the other winter cereals, this crop guarantees a higher starch and protein content in the biomass than triticale, and this results in potential benefits for anaerobic digestion feeding and methane production [40]. Furthermore, the whole-crop wheat harvested at an early dough stage showed a higher ADL content than barley and triticale, with the exception of 2014–15 growing season. According to Weinberg et al. [41], ADL is the parameter that has the most impact on NDF digestibility and, in the present study, it probably reduced the advantages of the higher starch content in terms of BMP. Rincon et al. [42] also reported that the BMP of whole-crop wheat was slightly lower at the early dough stage than at the milk stage (346 vs 360 $Nm^3CH_4$ t $VS^{-1}$). Since ADL increased with the ripening stages [31,43], the choice of the correct harvesting time for each genotype and environment is fundamental to maximize the production of high-quality forage from whole-crop wheat and, at the same time, to increase the grain-to-biomass ratio. In this context, the selection of genotypes with a high NDF digestibility and stay green, which are also related to the tolerance to foliar diseases, could make it possible to reach the later ripening stages without compromising the digestibility of the fiber [44]. The $F_1$ wheat hybrid cultivar, which has a greater vegetative vigor and high biomass production, may be able to provide genetic materials that could be suitable for such energetic uses in the near future [45]. Moreover, since Ronga et al. [31] reported that the nutritive value of whole-crop wheat is no different when harvested at heading or at the milk stage, due to the better NDF digestibility, an alternative approach could be to anticipate the wheat harvest, while maintaining a biomass advantage over triticale, to favor a higher productivity of the intercropping maize.

The $F_1$ hybrid barley resulted in the same DM biomass yield as triticale, except for in the 2015–16 growing season, while, since the BMP values were similar, the methane yield per area unit was never statistically different between these crops. Moreover, the whole barley crop resulted in a higher starch content than triticale, thus confirming the findings of Nadeau [39], who highlighted an overall higher organic matter digestibility for conventional barley cultivars harvested at the early dough stage. The chemical composition and in-vitro DM digestibility of the barley and wheat silages were similar, while the in-vivo DM digestibility has been reported to be higher in barley-fed cows [46]. Moreover, barley had a comparable methane yield per kg of DM to that of maize [25], thus confirming the potential advantage of using this crop as forage, above all, if the biomass yield of this crop can be increased. The use of $F_1$ hybrids instead of conventional cultivars led to higher whole-crop biomass yields of the winter barley for bioenergy production, as reported in the multi-location field experiment of Bernhard et al. [25], who stated that hybrids resulted in a higher plant height and ear dimension than pure lines. Preiti et al. [47] reported a higher biomass yield (+18%) of hybrids than conventional barley cultivars in Mediterranean environments because of a higher number of tillers and larger culms. In addition to the higher biomass yield, the barley hybrid also resulted in a higher stay green and DM digestibility than conventional cultivars [48], and these in turn led to a better methane yield per DM unit, but also reduced the risk of the deterioration of the silages and the storage nutrient losses that are typical of whole-crop barley silage, which may occur as a consequence of a low biomass compaction [49]. Furthermore, a better yield stability has

been reported for $F_1$ hybrids [27,50], which is of particular interest for marginal and drought-prone environments.

The present experiment underlines that hybrid barley can guarantee a methane yield that is comparable to that of triticale. Furthermore, the great advantage of using barley as a whole crop, instead of other small cereals, is connected to the potential of minimizing the yield penalty related to the late planting of the intercrop maize, since it is harvested 1–2 weeks before triticale or wheat. Several researchers [9,15,51–53] have stated that there is a potentially higher role for the energy yield of $C_4$ intercrop maize than that of $C_3$ winter cereal in intensified double-cropping systems in temperate areas. Thus, all the crop practices that are able to maximize the methane yield capacity of maize play fundamental roles in improving the energy efficiency of the overall system. The data collected in this trial established that the BMP of whole-crop maize (on average 339 $Nm^3CH_4$ t $VS^{-1}$) did not change when maize was cultivated as a sole-crop or an intercrop, even when considering different sowing timings, thereby confirming the findings of Strauß et al. [52] and Wannasek et al. [53]. Thus, maximization of the methane yield per unit area could be achieved by focusing on strategies that are able to optimize the biomass yield of maize. In the present study, the two main agronomic techniques adopted to optimize radiation interception in maize were considered, i.e., the positioning of the crop cycle, as a consequence of the sowing time, and the crop density.

It has been reported that when maize sowing is delayed in temperate growing areas, grain filling takes place in a period in which there is a progressive deterioration of the photo-thermal conditions for crop growth [54]. In addition to the optimized position of the crop cycle within the growing season, the results of the present study have also shown the advantage of applying a high plant density (HiD) for maize, compared to the standard one (StD), to obtain a high methane yield, which had previously been reported for both maize grain [24] and biomass [55]. The enhanced plant density increased the lead area index [56], thus leading to a positive result on the cumulative amount of intercepted incident photosynthetically active radiation and, consequently, on maize biomass production [57]. Moreover, the increase in plant density did not change the BMP of whole-crop maize, thus confirming previous finding [16,55].

The most interesting result of the present study is the interaction that was observed between the benefits, in terms of methane yield obtained through a higher plant density, and the timing of the sowing. In fact, a higher biomass yield, due to the adoption of a high plant density, was only observed for the maize sole-crop or when it was sown after barley as opposed to triticale and wheat. We speculate that the absence of a positive effect of HiD for the late maize planting (after triticale or wheat) is related to the lower availability of radiation for maize during crop development. Irmak and Djaman [58] found that the increase in the maize grain yield as a result of a higher plant density varied to a great extent, according to the planting date and years. Djaman et al. [59], who compared different planting densities and sowing times, reported that when the optimum density for each genotype is reached, increasing the density is associated with a decrease in the maize grain yield, since the radiation use efficiency decreases while the competition among plants for water and nutrient increases. Bonelli et al. [54], who explored a broad range of sowing dates in a temperate maize region, reported that the progressive reduction in radiation and temperature during the reproductive period when the sowing date was delayed made the source (supply of assimilates to grain) more limiting than the sink (demand for assimilates by grains) for maize growth. Thus, the authors stated that grain yield responses to increases in plant density cannot be expected to occur when the source capacity is the limiting factor (e.g., late sowing dates).

As far as the input requirement of the cropping systems is concerned, the higher maize plant density could lead to a higher use of nutrients and water, thus this agronomic solution is less suitable for fields characterized by lower soil fertility, or when the supply of nutrients through the fertilization is not adequate and in no irrigated or in less water availability conditions [24]. Notwithstanding the greater biomass and methane yield, this

greater use of agronomic inputs, also necessary to support the double crop system, could partially limit the environmental performance and the energy balance of the innovative cropping system [38]. Further studies focusing on the comparison of energy efficiency environmental and economic parameters are required to fully evaluate the beneficial of the proposed innovative cropping system.

## 5. Conclusions

The analysis of the different cropping systems has highlighted the importance of correctly choosing the double crop combination to maximize the biomass and methane yields, and of exploiting all the available yield-increasing strategies (double-cropping system and the high density sowing of maize). Choosing a winter crop, such as $F_1$ hybrid barley, which is able to combine a good biomass yield with an earlier harvest time, has proved to be a key factor in achieving this goal, since HiD led to significant yield increases over StD, albeit only when maize was sown early in the season as an intercrop. In short, this study has shown that an intensive high-population maize, with up to 10 plants per $m^{-2}$, can lead to a real yield enhancement of both the biomass and methane yield in a single crop or in double cropping systems when the most appropriate early sowing time is chosen. The combination of the $F_1$ barley hybrid with high-density maize as an intercrop resulted in the highest energy potential in the irrigated temperate growing areas. Instead, with the later sowing of maize, after the cultivation of triticale or wheat harvested as a whole crop for biomass, the lower amount of solar radiation available during the crop cycle reduced the advantage related to the application of a high maize plant density. In order to maximize the methane yield per hectare, the optimal management of the cereal cropping systems needs to be continuously investigated, taking into account the expected development of new hybrid genotypes for winter cereals and maize genotypes able to tolerate higher plant density.

**Author Contributions:** Conceptualization, M.B.; methodology, M.B. and E.D.; validation, M.B. and E.D.; formal analysis, M.B., M.S. and L.R.; investigation, M.B.; data curation, M.B.; writing—original draft preparation, M.B.; writing—review and editing, M.S., L.R. and A.R.; supervision, M.B., A.R. and E.D.; project administration, M.B., funding acquisition, M.B. and A.R. All authors have read and agreed to the published version of the manuscript.

**Funding:** This research was funded by Syngenta Italia spa, Milan, Italy.

**Institutional Review Board Statement:** Not applicable.

**Informed Consent Statement:** Not applicable.

**Data Availability Statement:** The data presented in this study are available on request from the corresponding author.

**Acknowledgments:** The authors would like to thank all the field and lab technicians who made a valuable contribution to the study and Marguerite Jones for the English editing. Thanks are also due to the farmers who hosted the experimental studies in their fields and collaborated closely with the present research team throughout the study.

**Conflicts of Interest:** The authors declare no conflict of interest. The funder played no role in the design of the study, in the collection, analyses or interpretation of the data, in the writing of the manuscript or in the decision to publish the results.

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
