# Peer review of "Biomass and Methane Production in Double Cereal Cropping Systems with Different Winter Cereal and Maize Plant Densities"

_agronomy, doi:10.3390/agronomy13020536_

Round 1

Reviewer 1 Report

- please reformulate the title, the word "precocity" is inappropriate

- the abstract section must improve according to the suggestion

- the introduction section must be improved with relevant references regarding the pretability of each analyzed species for methane production and the relation between quality and methane production

- in the results section- in my opinion, it is advisable to compare the second crop season with a normal crop season, maybe you should set another control/check

- please add some soil analysis for the experimental locations

- line 222- ANOVA showed a significant effect on or between species? please clarify...

- line 225- 30.6 and 22.1% than barley and triticale (reverse)

- line 227-228better use just Biomass Yield whiteout DM so as not to get confused with DM%

- line 257- please replace "maize sowing time" with single crop/ double crop

- please replace the table near the comments, and also finish the comments for one table and only then go to the next table with comments

-line 311- please rediscuss- there are some differences highlighted by Duncan test

- line 348- in the second experiment year the triticale has an increase value

- line 366-370- this reference is not relevant

- line 393- also this reference is not relevant

- line 411- it is not a successful comparison because the cereals are harvested in this period

- line 424-also, irrelevant reference (51)

- attention to references they are not numbered properly

-the Discussion section must improve with relevant references

- the Conclusion section is not well defined

Reviewer 2 Report

The paper "Biomass and Methane Production in Double Cereal Cropping Systems with Different Winter Cereal Precocity and Maize Plant Densities" submitted for review concerns the possibility of obtaining biomass from agricultural crops as a substrate for biogas plants. The research concerned new technologies of cultivation of various plants in the mix cropping system and their methane potential. The introduction provides an interesting and appropriate background for further empirical research. However, it could be updated, because it is based mainly on literature items from 2005-2017 and only one item from 2020 and 2019. The research methodology is appropriate. The results of the research are presented clearly, and their discussion is their interpretation in the context of research by other authors. A valuable aspect of the work are conclusions of a utilitarian nature. However, they should be separated from the discussion section as a separate conclusions section. Colorful graphics and photos of crops would make the work more attractive. However, this is not a substantive allegation. Therefore, I believe that this work will be of interest to many readers and I recommend it, with minor corrections, for publication in the journal Agronomy.

Reviewer 3 Report

This is an interesting paper studying the use of double cropping and narrow row maize in biomass production for anaerobic digestion. It is well written and documented.  A comment about the difference in inputs of the compared cropping practices would be useful. Although a full comparison is another paper some indication on the differences should be of interest.

One comment:

L 246-248  As for the  methane yield per hectare, wheat always resulted in the highest value: on average this crop had a 28% and 17% higher value than triticale and barley, respectively

Is this correct or should be barley and triticale respectively?

Round 2

Reviewer 1 Report

The author has made the suggested changes.